# Kombucha Reduces Hyperglycemia in Type 2 Diabetes of Mice by Regulating Gut Microbiota and Its Metabolites

**DOI:** 10.3390/foods11050754

**Published:** 2022-03-05

**Authors:** Suyun Xu, Yanping Wang, Jinju Wang, Weitao Geng

**Affiliations:** Key Laboratory of Food Nutrition and Safety, Ministry of Education, Food Engineering and Biotechnology Institute, Tianjin University of Science & Technology, Tianjin 300457, China; xusuyun1112@163.com (S.X.); ypwang40@163.com (Y.W.); bioorange@tust.edu.cn (J.W.)

**Keywords:** kombucha, type 2 diabetes, gut microbiota, SCFAs, intestinal barrier

## Abstract

Kombucha, which is rich in tea polyphenols and organic acid, is a kind of acidic tea soup beverage fermented by acetic acid bacteria, yeasts, lactic acid bacteria. Kombucha has been reported to possess anti-diabetic activity, but the underlying mechanism was not well understood. In this study, a high-fat, high-sugar diet combined with streptozotocin (STZ) injection was used to induce T2DM model in mice. After four weeks of kombucha intervention, the physiological and biochemical index were measured to determine the diabetes-related indicators. High-throughput sequencing technology was used to analyze the changes in gut microbiota from the feces. The results showed that four weeks of kombucha intervention increased the abundance of SCFAs-producing bacteria and reduced the abundance of gram-negative bacteria and pathogenic bacteria. The improvement in gut microbiota reduced the damage of intestinal barrier, thereby reducing the displacement of lipopolysaccharide (LPS) and inhibiting the occurrence of inflammation and insulin resistance in vivo. In addition, the increased levels of SCFAs-producing bacteria, and thus increasing the SCFAs, improved islet β cell function by promoting the secretion of gastrointestinal hormones (GLP-1/PYY). This study methodically uncovered the hypoglycemic mechanism of kombucha through gut microbiota intervention, and the result suggested that kombucha may be introduced as a new functional drink for T2DM prevention and treatment.

## 1. Introduction

Type 2 diabetes mellitus (T2DM) is a chronic metabolic disease caused by reduced insulin sensitivity of tissue cells and insufficiency of insulin secretion [1]. High blood glucose and an associated disorder in glucose and lipid metabolisms are the main diagnostic markers of diabetes [2,3]. With the improvement of people’s living conditions and diets, T2DM incidence continuously rises worldwide. According to a report issued by the World Health Organization, 5.4% of the world’s population is likely to suffer from T2DM in 2025 [4]. Therefore, T2DM has become a pressing worldwide public health problem.

Although the pathogenesis of T2DM is still not fully understood, gut microbiota dysbiosis is thought to be closely related to its development [5]. Many studies have found that gut microbiota affected the pathological status and physiological functions of the host through various manners [6,7]. It was shown that gut microbiota participated in the development of T2DM through three pathways, including the short-chain fatty acid pathway, endotoxin pathway and bile acid pathway [8,9]. For the endotoxin pathway, the disordered gut microbiota, including the increase in gram-negative bacteria or pathogenic bacteria, and the decrease in beneficial bacteria, would damage the intestinal mucosal barrier by increasing the content of intestinal lipopolysaccharides (LPS). Consequently, low-level inflammation caused by the damaged mucosal barrier will lead to damage of islet β cells, as well as systemic insulin resistance [10,11]. The short-chain fatty acids (SCFAs), which are continuously synthesized by some typical probiotics in the intestinal tract, could activate the intestinal L cells to secrete gastrointestinal hormones (i.e., GLP-1 or PYY) [12]. GLP-1 and PYY could regulate glucose metabolism and energy balance in the host [13]. However, the high-fat, high-sugar diet (HFHSD) could disturb the balance of the gut microbiota, thereby blocking the production of SCFAs and eventually causing T2DM [14].

At present, long-term medication and intervention are needed to manage T2DM. Therefore, natural anti-diabetic functional foods may be a necessary supplementary intervention strategy. An increasing number of studies have shown that fermented functional beverages, such as kombucha, kefir, and enzymes, have anti-diabetic effects [15,16,17]. Kombucha is a traditional fermented tea drink with a long history [18]. The drink is prepared with two components: sugar, tea and water are added as raw materials, while inoculum comprising mainly of acetic acid bacteria (such as *Komagataeibacter xylinus*, *Acetobacter malorum*, *Gluconacetobacterxylinus*, etc.), yeast (such as *Saccharomyces cerevisiae*, *Zygosaccharomycesbailii*, *Candidaparapsilosis*, etc.) and lactic acid bacteria (such as *Lactobacillus plantarum*, *Lactobacillus bulgaricus*, etc.) are added as fermentation bacteria [19]. The final product has a refreshing taste and is rich in active functional components such as tea polyphenols and organic acids, which are widely reported to possess a regulatory effect on gut microbiota [20,21]. Previous studies have proven that kombucha has a hypoglycemic effect on mice after measuring the physiological, biochemical and histological characteristics changes [22]. However, studies reporting the ameliorative effect of kombucha on T2DM via gut microbiota regulation have yet to be reported. To investigate the possible effect of kombucha on improving T2DM, HFHSD combined with streptozotocin (STZ) was used to induce T2DM mice in this study. The intestinal SCFAs, gut micobiota diversity, intestinal mucosal barrier and inflammatory status were systematically evaluated to uncover the ameliorative effect of Kombucha. This study may provide the theoretical basis for the development of health-promoting beverages in the treatment of T2DM.

## 2. Materials and Methods

### 2.1. Materials and Reagents

Green tea and sucrose were purchased from Hangzhou Ming-Hangzhou Tea Co., Ltd. (Hangzhou, China). *Komagataeibacter xylinus* 1512, *Zygosaccharomyces bailii* 1484 and *Lactobacillus plantarum* MA2 were isolated from traditional kombucha and stored in the China General Microbiological Culture Collection Center (accession number: CGMCC 16184, CGMCC 16183, CGMCC 3005, respectively). Streptozotocin (STZ) was purchased from Sigma (St. Louis, MO, USA). The normal diet and HFHSD were purchased from Sibeifu Experimental Animal Technology Co., Ltd. (Beijing, China). Aspartate aminotransferase (AST), alanine aminotransferase (ALT) and glycogen measurement kits were purchased from Tianjin Solomon Biotechnology Co., Ltd. (Tianjin, China). Interleukin-6 (IL-6), tumor necrosis factor-α (TNF-α), insulin (INS), lipopolysaccharide (LPS), Glucagon-like peptide-1 (GLP-1), and recombinant Peptide YY (PYY) kits were purchased from Nanjing Jiancheng Bioengineering Institute (Nanjing, China). Trizol, 2× Taq PCR Mix, HiFiScript cDNA Synthensis Kit and BeyoFastTM SYBR Green qPCR Mix were purchased from Hangzhou Baosai Biological Technology Co., Ltd. (Shanghai, China). The normal diet was made up of 55% corn flour, 25% bean cake, 15% wheat bran, 3% multivitamin and 2% salt [23]. HFHSD is formulated with 50% normal diet, 22% sugar, 16% lard, 9% casein, 0.5% cholesterol, 0.5% cholate and 2% gelatin [23].

### 2.2. Preparation of Tea Soup and Kombucha

The tea soup was prepared by boiling ten grams of green tea leaves in 1000 mL water for 5 min. The green tea leaves were then removed by filtering. The obtained tea soup was added with 80 g sucrose and sterilize at 90 °C for 10 min for further use.

For the preparation of the kombucha, the starters: *Komagataeibacter xylinus* 1512, *Zygosaccharomyces* 1484 and *Lactobacillus plantarum* MA2, were cultured to a concentration of 10^8^ CFU/mL at 30 °C. The 100 mL culture were then centrifuged at 6000 RPM to collect the cells. The collected cells were washed with PBS twice and resuspended in 10 mL for use as a starter in the future preparation of kombucha. The CFU of the starter was calculated by plate colony count before inoculation [24]. Then, the mixed starter made up of *Komagataeibacter xylinus* 1512, *Zygosaccharomyces* 1484 and *Lactobacillus plantarum* MA2 at a ratio of 2:1:1 was inoculated into the tea soup. The total number of inoculated cells was 4 × 10^9^ CFU. Then the tea soup was fermented statically at 30 °C for 7 days. At the end of the fermentation, the fermented tea soup was filtered and sterilized at 90 °C to obtain the kombucha.

### 2.3. Components Analysis

The acidity was determined according to the acid-base titration method described in GB 541334-2010. The total sugar was determined by the phenol sulfuric acid method described in GB/T5009.8-2003. The total phenol acid content was determined using a total phenol acid detection kit (Beijing Regen Biotechnology Co., Ltd., Beijing, China). The organic acids (formic acid, acetic acid, butyric acid, citric acid, lactic acid) were determined using high-performance liquid chromatography (HPLC) system (Essentia LC-10AT, Shimadzu, Japan) [22].

### 2.4. Animal Experimental Design and Sample Collection

All of the animal experiments in this study were conducted in accordance with the Declaration of Helsinki, and the protocol was approved by the Ethics Committee of Tianjin University of Science and Technology (approval number: TUST20190921). The specific pathogen-free (SPF) grade male Kunming mice (4 weeks, 22–24 g) were purchased from SPF (Beijing) Biotechnology Co., Ltd. (Beijing, China) with a production license number of SCXK (jing) 2019-0010. The experimental animals were free to drink and eat and were kept in a clean animal breeding room in Tianjin University of Science and Technology (temperature 22 ± 2 °C and humidity 55 ± 5% with12 h light/dark cycle). 

In this study, 60 Kunming mice were randomly divided into 5 groups (12 mice in each group, *n* = 12):the normal control group (NC, normal diet and gavage administration with 11.1 mL/kg·bw saline), the T2DM group (DC, HFHSD and gavage administration with 11.1 mL/kg·bw saline), the positive medicinal control group (MET, HFHSD and gavage administration with 0.13 g/kg·bw metformin), the kombucha group (KT, HFHSD and gavage administration with 11.1 mL/kg·bw kombucha), and the tea soup group (TS, HFHSD and gavage administration with 11.1 mL/kg·bw tea soup). 

The experiments were comprised of a modeling period and a treatment period as shown in Figure 1. The mice feed with HFHSD was intraperitoneally injected with STZ (50 mg/kg·bw) 4 times (day 28, day 31, day 34 and day 37) to induce T2DM. If the concentration of FBG on day 40 was above 11.1 mmol/L, the mice would be included in the diabetes model [25]. Metformin, kombucha, tea soup and saline were gavage administrated for 28 days according to the various groups of mice. During the experiment, the diet intake, body weight and fasting blood glucose (FBG) were measured every week. At the end of the experiment, samples were collected. The feces were collected and immediately stored at −80 °C. All the mice were fasted for 12 h, and the blood was collected and stored at −80 °C. The liver, pancreas and colon tissues were taken, washed with saline and weighed. Part of the tissues was fixed in 4% paraformaldehyde solution for histopathological experiments. The other part was stored at −80 °C for further analysis.

### 2.5. Oral Glucose Tolerance Test (OGTT)

The oral glucose tolerance test was conducted on day 67. After fasting for 12 h, each mouse was orally administered 2.0 g/kg·bw glucose. Then, the blood samples were collected from the tail vein after 0, 30, 60, 90 and 120 min for the measurement of blood glucose level. The blood glucose level curve versus time was drawn. The area under the curve (AUC) was calculated by using Formula (1) [26].
(1)$$\mathrm{AUC}=\frac{(0\;h\;\mathrm{blood}\;\mathrm{glucose}+0.5\;h\;\mathrm{blood}\;\mathrm{glucose})\times 0.5}{2}+\frac{(1\;h\;\mathrm{blood}\;\mathrm{glucose}+0.5\;h\;\mathrm{blood}\;\mathrm{glucose})\times 0.5}{2}+\frac{(2\;h\;\mathrm{blood}\;\mathrm{glucose}+1\;h\;\mathrm{blood}\;\mathrm{glucose})\times 1}{2}$$

### 2.6. Determination of Organ Index and Biochemical Indexes

After 4 weeks of gavage administration (day 68), all mice were fasted for 12 h and weighed. The mice were sacrificed by cervical vertebrae removal. The liver and pancreas were taken and weighed to calculate the organ index by using Formula (2).
(2)Organ index(%)=Organ wet weight÷weight×100

The ALT and AST in the serum and glycogen in the liver tissue were determined with colorimetric assay from biochemical kits. The GLP-1, PYY, TNF-α, IL-6, LPS and FINS in serum were detected using an ELISA kit. The insulin and the fasting blood glucose (FBG) concentration were determined and used to calculate the homeostasis model insulin resistance index (HOMA-IR) and the homeostasis model insulin secretion index (HOMA-β) using Formulas (3) and (4) [27].
(3)HOMA−IR=FBG×FIns÷22.5
(4)HOMA−β=20× Fins/(FBG−3.5)

### 2.7. Histopathological Evaluation

The liver, pancreas and colon of the mice were collected for the preparation of paraffin tissue sections. The morphologies of the tissues were observed using a microscope (Nikon, Japan) after staining with hematoxylin and eosin (H&E) [28]. The histopathological score of the tissues was obtained according to the standard in Appendix A.

### 2.8. Gut Microbiota Analysis

The collected feces were sent to Guangzhou Kitdio Biotechnology Co., Ltd. (Guangzhou, China) for 16S rDNA high-throughput sequencing. The sequencing was conducted by using an IlluminaMiSeq platform. After sequencing, the effective sequences were obtained from the original data by filtering. The effective sequences were classified into multiple operational taxonomic units (OUTs) at a similarity threshold of 97%. The species information OUTs were annotated using Ribosomal database project (RDP) databases and the reliability was more than 80%. Finally, QIIME software was used to analyze the sequencing results [29,30]. The raw data were deposited in the NCBI sequence read archive (SRA) database (accession number: PRJNA778664).

### 2.9. Real-Time Quantitative PCR (RT-qPCR) Analysis

Total RNA was extracted from the colon tissue by using TRIzol reagent. The quality and concentration were detected by using agarose gel electrophoresis. RNA with high purity and good integrity was reversed transcribed to cDNA by using a reverse transcription kit (Appendix A). The obtained cDNA was used as the template for RT-qPCR analysis (Appendix A). The RT-qPCR primers used in this study are listed in Appendix A. The relative difference in mRNA transcription levels of the different groups was calculated using 2^−ΔΔCt^ method [31].

### 2.10. The Short-Chain Fatty Acid (SCFA) in Fecal Samples

The content of SCFAs (formic acid, acetic acid and butyric acid) in the feces was determined by HPLC (Essentia LC-10AT, Shimadzu, Japan) [22]. The mobile phase was 95% phosphoric acid and the flow rate was 0.1 mL/min. Column temperature was 35 °C. The detection wavelength was 210 nm. The samples were prepared by using 0.2 g of the collected feces and dispersing them fully in 0.8 mL of phosphoric acid solution (0.1%, *v*/*v*). The supernatants of the feces suspension were obtained by 13,000 r/min of centrifugation at 4 °C for 10 min, followed by filtration with 0.22 μm microporous cellulose membrane. The standard curves of each type of SCFA were prepared by using the standard solution comprising of different concentrations (0.08 g/L, 0.16 g/L, 0.32 g/L, 0.64 g/L, 1.28 g/L). Following this, the types and concentrations of the SCFAs in the samples were detected and analyzed according to the standard curve.

### 2.11. Statistical Analysis

All the experiments were repeated three times. The data were presented as means ± SD. Statistical analysis was performed by using Statistical Product and Service Solutions 22.0 (SPSS 22.0). Differences were compared using one-way analysis of variance. A value of *p* < 0.05 or *p* < 0.01 was regarded as having significant difference or extremely significant difference, respectively.

## 3. Results and Discussion

### 3.1. The Fermentation Provided Kombucha with Different Components Compared with Tea Soup

The main effective components (such as organic acids and phenolic) of kombucha were reported to have anti-inflammatory, anti-bacterial, hypoglycemic and regulatory effects on gut microbiota [32,33]. By comparing the components in tea soup and kombucha, it was determined that these effective components were produced by microbial transformation through the fermentation process during the preparation of kombucha. As shown in Table 1, the contents of total acid (8.6 mg/mL) and total phenol acid (1.39 mg/mL) in kombucha were significantly higher than those in tea soup (*p* < 0.01), while the content of total sugar (18.73 mg/mL) was significantly lower (*p* < 0.01). The results indicated that the increase in active substances (such as phenols and organic acids) in kombucha may have been produced through microbial fermentation of the carbon source and other substances found in the tea broth. Previous studies have shown that yeast, acetic acid bacteria and lactic acid bacteria can increase the total acidic content in fermentation broth through the EMP pathway and anaerobic fermentation [19,34,35]. At the same time, the hydrolase produced by microorganisms during the fermentation process can degrade insoluble phenols in tea soup, thereby releasing phenolic acids which improve the content and activity of total phenols [35]. This suggested that microbial transformation during the fermentation process is essential to the function of kombucha. In addition, among the organic acids showing increased content after fermentation, the content of acetic acid is the highest (3.03 mg/mL), followed by citric acid, butyric acid, lactic acid and formic acid. The acetic acid and citric acid are not only used as sour agents to improve the taste of the product, but also to prevent the growth of miscellaneous bacteria [36]. The butyric acid is reported to demonstrate effects of anti-inflammatory, hypoglycemic, and gut microbiota regulating [14]. 

### 3.2. Effects of Kombucha on Fasting Blood Glucose, Food Intake, Body Weight, Glucose Tolerance, Insulin and Glycogen Contents in T2DM Mice

This study evaluated the effect of kombucha on T2DM in mice by monitoring the fasting blood glucose (FBG), food intake and body weight of mice during the experiment. The above values were measured before STZ induction (day 28), after STZ induction (day 37) and during the 4 weeks of administration intervention (day 40, day 47, day 54, Day 61 and day 68). First, the levels of FBG were compared (as shown in Figure 2A). After 28 days of HFHSD feeding, the FBG of mice was about 7.02 mmol/L, which was 1.25 times higher than that of the NC group (5.61 mmol/L, *p* < 0.05). It indicated that the 28 days HFHSD can increase blood glucose in mice without entering into the state of diabetes. However, after the injection of STZ, the FBG of mice in each group increased significantly to 20–22 mmol/L, which was 4 times higher than that in NC group (*p* < 0.01). This indicates that the combined induction of HFHSD and STZ leads to the mouse becoming a T2DM model. Subsequently, the food intake and body weight of the mice in each group during the intervention period were compared, as shown in Figure 2B,C. The results showed that, when compared with the NC group, the food intake (11.78 g/mouse/day, *p* < 0.01) in the T2DM group increased by 2.2 times after STZ intervention while the body weight decreased by 9.65% (44.78 g, *p* < 0.01). At the same time, there existed a negative correlation between food intake change and body weight change of T2DM mice, with an R value of −0.927 (Appendix A, Appendix A. *p* < 0.01). The above changes in food intake, body weight and FBG are typical characteristics of T2DM [37]. After 4 weeks of KT intervention, compared with DC group, the FBG (15.22 mmol/L, *p* < 0.01) and food intake (13.59 g/mouse/day, *p* < 0.01) of mice decreased significantly, and the body weight recovered significantly (48.21 g, *p* < 0.01). In the TS intervention group, FBG, food intake and body weight were 17.77 mmol/L, 15.2 g/mouse/day and 42.81 g/mouse/day, respectively. The results showed that compared with unfermented tea soup, kombucha tea can more effectively improve the hyperglycemic characteristics of T2DM mice. The polyphenols and organic acid active substances in kombucha may play a role in reducing blood glucose.

The regulatory effect of kombucha on blood glucose was evaluated by measuring the oral glucose tolerance test (OGTT), insulin resistance, and the function of islet β cells at Day 68. For the oral glucose tolerance test (Figure 2D), the blood glucose in NC group returned to normal level in 2 h, while the blood glucose in DC group was still maintained at a high level. Meanwhile, the AUC value in DC group was nearly three times higher than the NC group (Figure 2E). After 4 weeks of intervention, the blood glucose in OGTT was returned more quickly in the MET, KT and TS groups than that in the DC group while the AUC values in the MET, MT and TS groups were also significantly decreased by 38.11%, 31.21% and 14.72% compared with those in the model group (*p* < 0.01). The results indicated that the glucose metabolism in T2DM mice was seriously damaged. Kombucha can effectively improve the glucose metabolism in the T2DM mice. The insulin resistance was subsequently evaluated by the steady-state model. Figure 2F,G showed that the insulin and HOMA-IR in DC group were 2.3 and 6.73 times of NC group, respectively; this depicted the severity of insulin resistance in the DC group. Compared with the DC group, the insulin resistance of the mice in the MET and KT groups was reduced by 50.83% and 42.15% respectively (*p* < 0.01). HOMA-β is an index used to evaluate the function of islet β cells [25]. It can be seen from Figure 2H that the insulin secretion index of islet β cells in T2DM mice is only 20.02% of the values in the NC group. After 4 weeks of intervention, the function of pancreatic islet β cells in the MET and KT groups were significantly increased by 1.37 and 1.31 times respectively (*p* < 0.01). The results indicated that kombucha can effectively alleviate insulin resistance and islet β-cell dysfunction in T2DM mice. Previous studies have shown that islet β-cell dysfunction in T2DM mice can lead to abnormal glucose metabolism, including glycogen synthesis [38]. It can be seen from Figure 2I that compared with the NC group (7.89 mg/mL), the liver glycogen content in the DC group (4.11 mg/mL) was significantly decreased (*p* < 0.01). After 4 weeks of intervention, the glycogen contents in MET (7.12 mg/mL, *p* < 0.01), KT (6.97 mg/mL, *p* < 0.01) and TS (4.66 mg/mL, *p* < 0.05) were significantly increased. The results hence showed that kombucha could effectively improve the liver glucose metabolism disorder in diabetic mice by enhancing glycogen synthesis. 

### 3.3. Kombucha Improved the Damaged Liver and Islet Tissue in T2DM Mice

Tissue sections of the liver and pancreatic islet, and hypohepatia index of blood tests were used to evaluate the effect of kombucha on improving the tissue damage in T2DM mice (Figure 3A–F). Through morphology observation of liver tissue, it was found that the liver tissue in DC group had liver cell edema, larger volume, disordered structure and local inflammatory cell infiltration (Figure 3A). The AST and ALT activities of the DC group increased by 1.5 times and 2.6 times compared with that in the NC group respectively (*p* < 0.01, Figure 3C,D). This implied that the liver function of the mice in the DC group was dysfunctional. After 4 weeks of intervention, the infiltration and necrosis of inflammatory cells in the liver of mice in KT group were significantly reduced, indicating that kombucha has a protective effect on liver tissue. The serum AST contents of mice in the KT and TS groups were significantly decreased, reduced by 17.3% and 10.7%, respectively (*p* < 0.01, *p* < 0.05), while the ALT contents were decreased by 25.4% and 12.14%, respectively (*p* < 0.01, *p* < 0.05). The results indicated that kombucha can effectively improve the damage of liver function. For the organ index, the liver coefficient in the DC group was 1.56 times higher than that in the NC group (Figure 3E, *p* < 0.01), indicating that severe hepatomegaly had occurred in the DC group mice which was induced by high fat, high glucose and STZ drugs. However, after the intervention of kombucha, the liver value of the KT group was significantly lower than that of the DC group (*p* < 0.01), which again proved that kombucha is able to protect T2DM mice from liver injury. 

Pancreatic islet injury is one of the mechanisms leading to the pathogenesis of diabetes [39]. It can be seen from Figure 3B that the islets in the DC group were severely damaged. The islet cells were significantly shrunk, as shown by disordered shape, disappearing outline and blurred peripheral boundary. After 4 weeks of intervention, the number of islet cells in the KT group increased and the arrangement became more uniformed, and the boundary between the islet and exocrine part was relatively clear. The results indicated that the kombucha could significantly repair islet injury. In addition, the pancreatic index in the DC group decreased by 29.6% compared with that in the NC group (Figure 4F, *p* < 0.01). After 4 weeks of intervention, the pancreatic index of the KT and TS groups increased by 1.26 and 1.1 times compared with the DC group (*p* < 0.01, *p* > 0.05), respectively. The results indicated that kombucha can also effectively protect the pancreas. 

### 3.4. Kombucha Changed the Structure of the Gut Microbiota in T2DM Mice

In recent years, a large number of studies have confirmed that gut microbiota is closely related to T2DM [40,41]. What is more, the symptoms of T2DM could be effectively alleviated by changing the gut microbiota [8]. Therefore, this study explored the hypoglycemic mechanism of kombucha from the perspective of gut microbiota.

To determine the α diversity, several indexes can be used. Chao1 index and ACE index reflected the richness of the community, while Shannon index and Simpson index reflected the diversity of the community [1]. The richness and diversity index in DC group were significantly decreased (*p* < 0.05). After the 4 weeks of intervention, the richness and diversity index in MET and KT group were significantly increased (*p* < 0.01, *p* < 0.05) (Appendix A).

A large number of studies have shown that HFHSD can cause gut microbiota disorder by changing the taxonomic composition of gut microbiota [42,43]. It is therefore important that the regulation effect of the kombucha on the taxonomic composition was investigated. The changes at the phylum level were shown in Figure 4A. Compared with the NC group, the proteobacteria and bacteroides in the DC group increased by 2.2 times and 1.2 times, while Firmicutes decreased by 21.7%. The change in gut microbiota is also a feature of T2DM [1]. Proteobacteria are gram-negative and often believed to include a variety of pathogens, including *Salmonella*, *Vibrio cholerae*, *Shigella*, etc. The increase in Proteobacteria is usually considered as a diagnostic marker for gut ecological disorders and disease risks [44]. The members of Firmicutes are mostly gram-positive bacteria. Firmicutes are usually considered to be anti-inflammatory bacteria because of their production of SCFA metabolites [45]. After 4 weeks of kombucha intervention, Firmicutes increased by 1.26 times and Proteobacteria decreased by 59% compared with the DC group. The taxonomic composition of the gut microbiota in the KT group was altered to resemble the normal group.

Figure 4C showed the cluster analysis of the top 20 genera. The compositions in the heat map are clustered according to their similarity. It can be seen from Figure 4C that the 20 genera could be divided into two categories, including SCFAs-producing bacteria (Figure 4D) and LPS-producing Gram-negative pro-inflammatory bacteria (Figure 4E). Compared with the NC group, the abundance of *Lactobacillus*, *Butyricicoccus*, *Lachnospiraceae_NK4A136_group* in the DC group were significantly decreased (*p* < 0.01, Figure 4C, D). After the intervention of kombucha, the relative abundance of these bacteria was significantly increased (*p* < 0.01). The abundance of *Bifidobacterium* was not significantly different between NC group and DC group, but significantly increased in KT group and TS group (*p* < 0.01, Figure 4C,D). A number of studies have shown that the abundance of *Lactobacillus*, *Bifidobacterium*, *Butyricicoccus*, and *Lachnospira* were significantly positively correlated with SCFAs content in the intestine [46,47]. In particular, the acetic acid and butyric acid produced by *Bifidobacterium* and *Butyricicoccus* had the effects of inhibiting the growth of pathogenic bacteria, controlling appetite to regulate energy balance, repairing intestinal mucosal injury, and slowing intestinal inflammation [48]. Therefore, the enrichment effect of kombucha on SCFAs-producing bacteria may be one of the explanations for its health benefitting actions. It can be seen from Figure 4E that compared with that in the NC group, the abundances of *Desulfovibrio*, *Escherichia-Shigella*, and *Bacteroidetes* in the DC group were significantly increased by 2.82, 4.9, and 1.8 times (*p* < 0.01), respectively. The abundances returned to a normal level after the intervention of kombucha. *Desulfovibrio* and *Escherichia-Shigella* are both Gram-negative bacteria. Many of them contain capsules and microcapsules, and the LPS produced can cause severe inflammation, leading to enteritis and diabetes [49,50]. Bacteroidetes are also Gram-negative bacteria, which contain most pro-inflammatory bacteria such as *Bacteroides faecalis*, *Bacteroides fragilis* [51]. Many studies have shown that tea polyphenols were closely related to gut microbiota [29,44]. Tea polyphenols, which are not easily and directly absorbed by the intestine, can promote the growth of SCFAs bacteria and inhibit harmful bacteria in the intestine [52]. At the same time, organic acids can maintain intestinal pH balance and promote the growth of probiotics [47]. The richness in active ingredients such as tea polyphenols and organic acids of kombucha may be the main reason for improving gut microbiota disorders.

A number of studies have shown that HFHSD led to gut microbiota imbalance and the increase of the proportion of Gram-negative bacteria [10,40]. An unbalanced gut microbiota can promote the increase of intestinal endotoxin, cause intestinal barrier damage and metabolic endotoxemia [53,54]. In this study, the proportion of gut microbiota G+/G− bacteria were analyzed by Bugbase tool. It can be seen from Figure 4F that the proportion of G− bacteria in the DC group was significantly higher than that in the NC group (57.71% vs. 41.52%, *p* < 0.01). After four weeks of kombucha intervention, the proportion of G− bacteria in the KT group was significantly lower than that in the DC group by 22.05%, indicating that kombucha had a regulating effect on the unbalanced gut microbiota in T2DM mice.

### 3.5. Kombucha Regulated the Intestinal Metabolites in T2DM Mice

SCFAs are the main products of dietary carbohydrates fermentation by intestinal microorganisms, which include formic acid, acetic acid and butyric acid [55]. The content of SCFAs in mouse feces, as determined by HPLC, was shown in Figure 5A–C. It can be seen that compared with the NC group, the contents of formic acid, acetic acid and butyric acid in the DC group were significantly decreased by 30.58% (*p* < 0.05), 64.27% (*p* < 0.01), 67.74% (*p* < 0.01), respectively. After four weeks of administering kombucha and tea soup, acetic acid was significantly increased by 2.2 and 1.45 times compared with the DC group (*p* < 0.01, *p* < 0.05). The content of formic acid did not change significantly in the different groups (*p* > 0.05). It is hence suggested that kombucha can effectively improve the content of acetic acid and butyric acid. Since kombucha contains organic acids such as butyric acid and acetic acid, the mice in the KT group also saw a promoted growth of SCFAs-producing bacteria, so these may be the reason for the higher SCFAs content in the KT group.

Considering that kombucha is able to regulate SCFAs, we further explored the GPR41/GPR43 mRNA expression and gastrointestinal hormones (GLP-1, PYY) secretion by using RT-PCR and ELISA, respectively. SCFAs can be recognized by two specific short-chain fatty acid receptors (GPR41 and GPR43), located on the intestinal L cells [8,13]. As shown in Figure 5D, compared with the NC group, the GPR41 and GPR43 mRNA expression levels in the DC group were significantly decreased by 45.65% and 39.5%, respectively (*p* < 0.01). After gavage with kombucha and tea soup for four weeks, compared with the DC group, their mRNA expression levels were significantly increased. This result indicated that the regulation of kombucha was achieved by activating GPR43/GPR41. The activation of GPR43/GPR41 could promote the secretion of intestinal proinsulin (GLP-1) and gastrointestinal hormone peptide (PYY), which could regulate glucose metabolism and protect islet βcells by promoting insulin secretion and glycogen synthesis [56]. The contents of serum gastrointestinal hormones were shown in Figure 5E. It showed that, compared with the NC group, the GLP-1 and PYY content in DC group were decreased by 31.11% and 39.13%, respectively (*p* < 0.01). After four weeks of kombucha administration, the content of GLP-1 and PYY were significantly increased to 8.4 pmol/L (*p* < 0.01) and 8.8 pmol/L (*p* < 0.05), respectively. The results indicated that kombucha may play an anti-diabetic role by affecting the gut microbiota-SCFAs-GPRs pathway.

### 3.6. Kombucha Improved the Intestinal Permeability in T2DM Mice

In T2DM mice, the disordered gut microbiota would cause intestinal inflammation and damage the intestinal mucosal barrier, which would then lead to systemic chronic inflammation through LPS displacement. The long period of systemic chronic inflammation will ultimately lead to insulin resistance and islet β cell function impairment [10,57,58]. Noting that gut microbiota is able to improve the intestinal condition, the alleviating effect on the intestinal mucosal barrier of kombucha were evaluated by detecting the expression level of colonic inflammatory factor mRNA, colonic histological feature and the LPS content in the blood.

Intestinal inflammation is closely related to gut microbiota imbalance [45]. The alleviating effect of kombucha on the colon inflammatory was determined. As shown in Figure 6A, the relative expression levels of IL-1β, IL-6 and TFN-α in DC group were 3.2, 4.2 and 2.6 times higher than those in NC group (*p* < 0.01). The IL-1β, IL-6 and TFN-α mRNA expression levels were significantly decreased (*p* < 0.01, *p* < 0.05) after 4 weeks of kombucha and tea soup administration, demonstrating that they possess an amelioration effect on intestinal inflammation. This may be due to the inhibitive effect on LPS-producing bacteria and the promoting effect on SCFA production by kombucha.

The intestinal barrier damage caused by gut microbiota disturbance was evaluated using H&E staining (Figure 6B). In the NC group, there is a complete colonic epithelium structure with tightly arranged villi structure, rich in goblet cells, uniform mucosal and muscle layers, complete crypts and glands, and no inflammatory cell infiltration. On the contrary, in the DC group, the mucosa tissues were severely damaged including a large number of infiltrated inflammatory cells, intermittently enlarged intestinal villi cells, disappearing goblet cells and crypts. On the other hand, the tissues in KT and MET showed good appearance, similar to the NC group. The tissue damage score (Appendix A) showed that, compared with the NC group, the tissue damage score of the DC group was significantly increased (*p* < 0.01). While the tissue damage score in the KT group was significantly decreased (score 3.52, *p* < 0.01), indicating the colonic injury could be restored by kombucha.

To clarify the protective mechanism of kombucha against intestinal barrier in T2DM mice, the expression levels of tight junction proteins (ZO-1, Claudin-1, Occludin) and mucin (Muc2) in the colon were investigated. As shown in Figure 6C, the ZO-1, Claudin-1, Occludin and Muc2 mRNA expression levels as in DC group were significantly decreased compared to those in NC group, by 61.51%, 66.72%, 64.21% and 75.01%, respectively (*p* < 0.01). Previous studies have also reported that the expression levels of tight junction proteins Occludin and Claudin-1 were downregulated by inflammatory factors and intestinal LPS through the NF-κB pathway [59,60]. After treating with kombucha and tea soup, the relative expression levels of the tight junction protein and mucin mRNA were significantly increased (*p* < 0.01, *p* < 0.05). The results suggested that kombucha may improve intestinal barrier injury by regulating the expression of tight junction protein and mucin-related genes. It is well known that SCFAs and polyphenols can induce secretion of MUC2 in goblet cells thereby protecting the intestinal barrier damage [61]. It can be concluded that kombucha regulated gut microbiota disorders and reduced intestinal inflammation to improve intestinal mechanical damage.

The contents of LPS, TNF-α and IL-6 in serum of mice were determined by evaluating the LPS displacement caused by the intestinal barrier damage. As shown in Figure 6D,E, compared with the NC group, the serum LPS, TNF-α and IL-6 in DC mice were significantly increased by 1.51, 1.77 and 2.05 times, respectively (*p* < 0.01), indicating that LPS displacement occurred in T2DM mice. After 4 weeks of intervention, the contents of LPS, TNF-a and IL-6 in KT group were significantly lower than those in DC group by 21.9%, 31.1% and 38.64%, respectively (*p* < 0.01). Compared with the DC group, LPS, TNF-a and IL-6 in the TS group were decreased by 5.6% (*p* > 0.05), 11.08% (*p* < 0.01) and 11.29% (*p* < 0.01), respectively. The results showed that the intestinal permeability was increased due to the damage of the intestinal mucosa, resulting in LPS displacement. Kombucha could improve the systemic inflammation by alleviating intestinal mucosal damage. In order to further verify the correlation between LPS and inflammatory factors, the Pearson correlation analysis was performed. The results in Figure 6G,H and Appendix A showed that LPS was positively correlated with TFN-α and IL-6, and the R values were 0.974 and 0.973, respectively. It indicated that the LPS from the intestine caused an increase in TNF-α and IL-6 content in the serum.

### 3.7. Correlation Analysis of Gut Microbiota and Biochemical Indicators

In order to further explore the relationship between gut microbiota and host biological indicators, Pearson correlation analysis was used to evaluate the correlation between the physical and chemical indicators of mice and intestinal microorganisms with relative abundance of TOP20 at genus level. The results were shown in Figure 7. *Butyricicoccus*, *Lachnospiraceae_NK4A136 group*, *Lactobacillus*, *Alloprevotella*, *Bifidobacterium* were negatively correlated with FBG, Glycogen, Glucose-AUC, HOMA-IR, LPS, TFN-α, and IL-6. However, they were positively correlated with HOMA-β and GLP-1, and most of these bacteria were SCFA-producing bacteria. The results confirmed that SCFA-producing bacteria possess biological functions to regulate glucose metabolism, elicit anti-inflammatory effect, and protect the function of islet cells. On the contrary, *Escherichia-Shigella*, *Desulfovibrio*, *Bacteroides*, *Corynebacterium_1* and *Anaerovorax* were positively correlated with FBG, Glycogen, Glucose-AUC, HOMA-IR, LPS, TFN-α and IL-6, and negatively correlated with HOMA-β and GLP-1. The results indicated that these microorganisms may cause body inflammation and induce insulin resistance. It was also proved that the gut microbiota is an effective intervention target for the regulation of the biochemical indexes.

## 4. Conclusions

This study proved that kombucha has an obvious anti-hyperglycemic effect by regulating the gut microbiota structure. The administration of kombucha could decrease the abundance of G− and pathogenic bacteria and increase the SCFAs-producing bacteria in the T2DM mice. The low abundance of G− and pathogenic bacteria helped to reduce the LPS content in the blood, thereby alleviating the systematic inflammation and insulin resistance. The increase in SCFA producing bacteria could improve the intestinal barrier damage by inducing the expression of tight junction proteins, which prevent LPS displacement in T2DM mice. In addition, the SCFA also promoted the secretion of gastrointestinal hormones (GLP-1 and PYY) which then improved the pancreatic β function that plays a role in regulating blood sugar. This study systematically revealed the hypoglycemic mechanism of kombucha from the perspective of the gut microbiota. The result of this study opened up a new possibility of effectively preventing and treating diabetes by intervening in the diet of patients, and also paved the way for improving the market value of kombucha.

## Figures and Tables

**Figure 1 foods-11-00754-f001:**
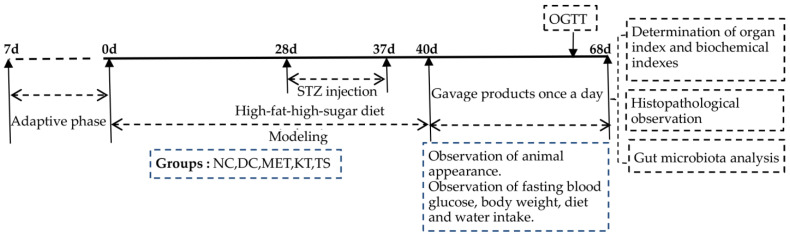
Schedule of the experiment.

**Figure 2 foods-11-00754-f002:**
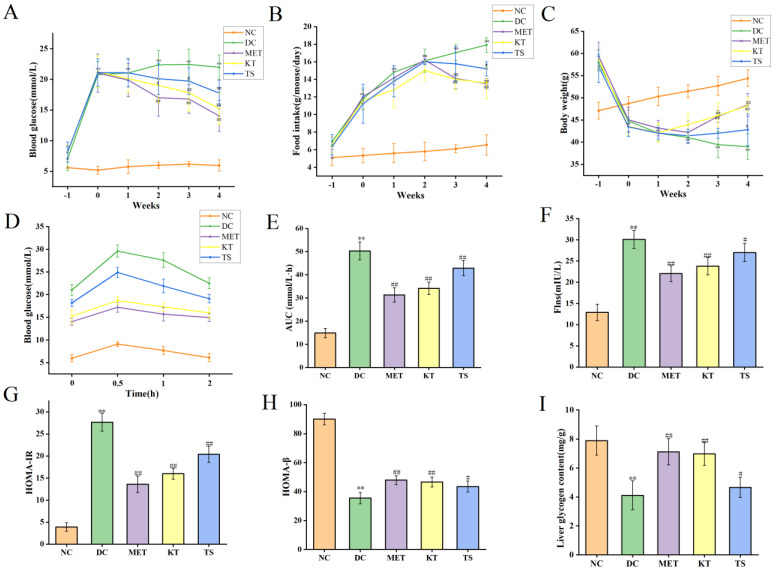
Effects of Kombucha on fasting blood glucose (FBG), food intake, body weight, glucose tolerance, insulin and glycogen contents in T2DM mice. (**A**) FBG; (**B**) Food intake; (**C**) Body weight; (**D**) Blood glucose levels of the oral glucose tolerance test (OGTT); (**E**) Area under the curve (AUC) of the OGTT; (**F**) *Flns* content in serum; (**G**) Homeostasis model assessment of insulin resistance index (HOMA-IR); (**H**) Homeostasis model insulin secretion index (HOMA-β); (**I**) Liver glycogen level. ** *p* < 0.01 vs. normal; # *p* < 0.05, ## *p* < 0.01 vs. model.

**Figure 3 foods-11-00754-f003:**
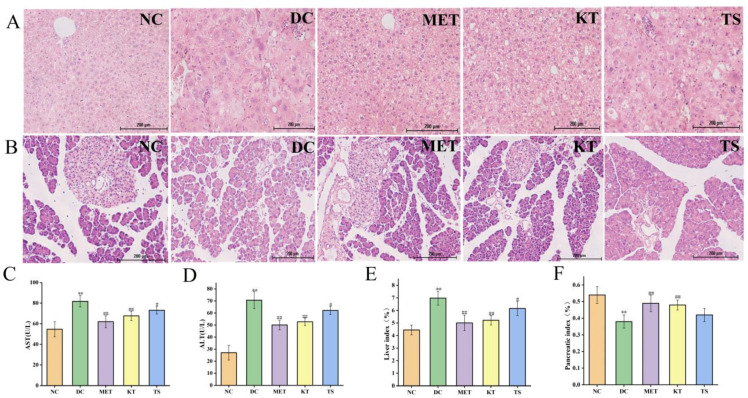
Effects of kombucha on liver and islet tissue injury in T2DM mice. (**A**,**B**) Histological examination of liver and islet slices with H&E staining (scale bars: 200 μm); (**C**,**D**) AST and ALT content in liver; (**E**,**F**) Liver and pancreatic index. ** *p* < 0.01 vs. normal; # *p* < 0.05, ## *p* < 0.01 vs. model.

**Figure 4 foods-11-00754-f004:**
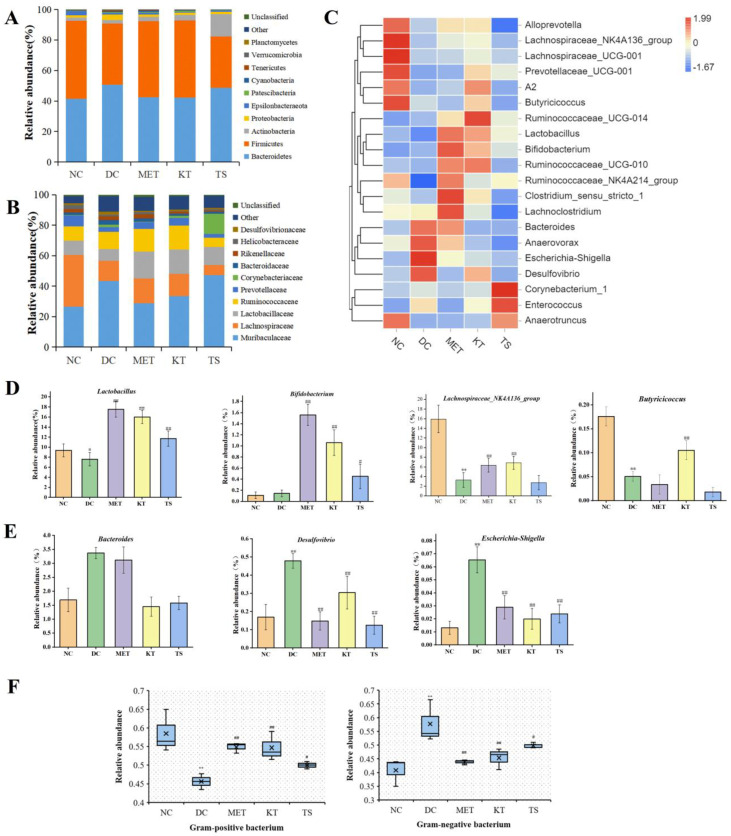
Effect of kombucha on the gut microbiota composition in T2DM mice. (**A**) Relative abundance at phylum level; (**B**) Relative abundance at family level; (**C**) Relative abundance at genus levels; (**D**) Relative abundance of SCFAs-producing bacteria (*Lactobacillus*, *Bifidobacterium*, *Butyricicoccus*, *Lachnospiraceae_NK4A136_group*); (**E**) Relative abundance of proinflammatory bacteria (*Desulfovibrio*, *Escherichia-Shigella*, and *Bacteroidetes*); (**F**) Relative abundance of gram-positive bacteria and gram-negative bacteria. * *p* < 0.05, ** *p* < 0.01 vs. Normal; # *p* < 0.05, ## *p* < 0.01 vs. Model.

**Figure 5 foods-11-00754-f005:**
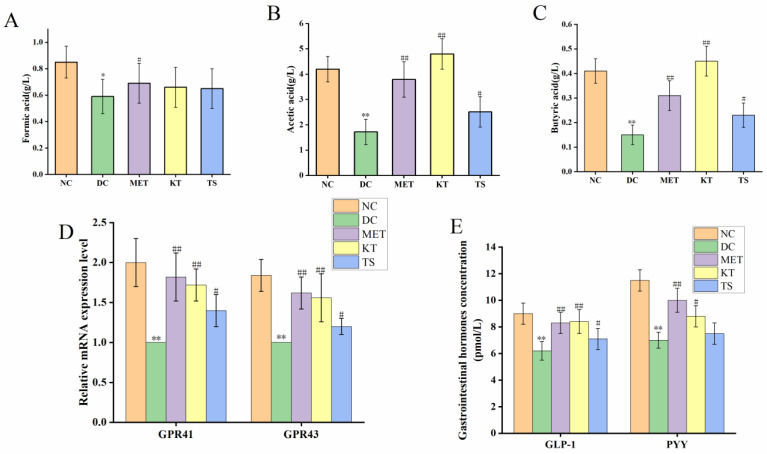
Kombucha regulated the intestinal metabolites in T2DM mice. (**A**–**C**) The content of SCFAs (formic acid, acetic acid, butyric acid) in the feces; (**D**) The mRNA levels of GPR41 and GPR43 in the colon tissue; (**E**) The levels of GLP-1 and PYY in the serum. * *p* < 0.05, ** *p* < 0.01 vs. Normal; # *p* < 0.05, ## *p* < 0.01 vs. Model.

**Figure 6 foods-11-00754-f006:**
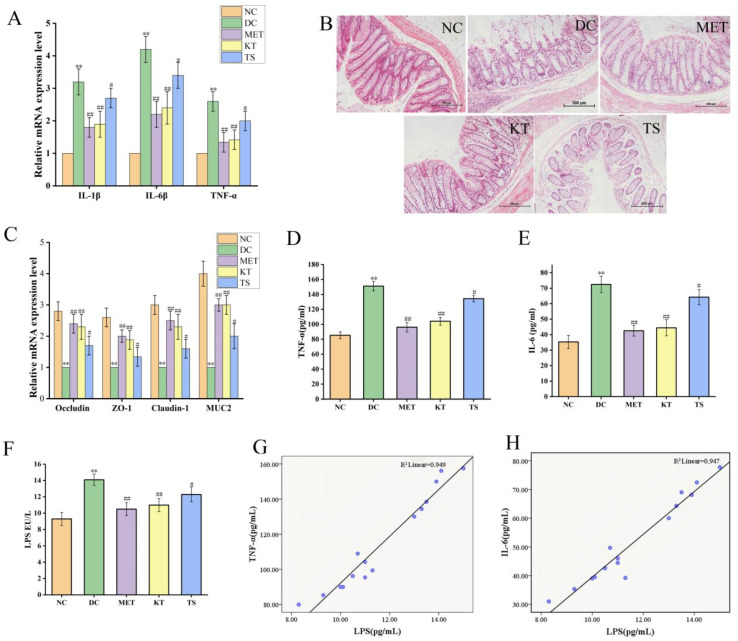
Kombucha improved the intestinal permeability in T2DM mice. (**A**) Relative gene expression of IL-1β, IL-6 and TNF-α in the colon; (**B**) Histological examination of colon with H&E staining (scale bars: 200 μm); (**C**) The mRNA levels of tight junction protein (TJP) Occludin, ZO-1, Claudin-1, and Muc2 in the colon; (**D**–**F**) The levels of TFN-α, IL-6 and LPS in the serum. (**G**,**H**) Correlation analysis of LPS with IL-6 and TNF-α. ** *p* < 0.01 vs. Normal; # *p* < 0.05, ## *p* < 0.01 vs. Model.

**Figure 7 foods-11-00754-f007:**
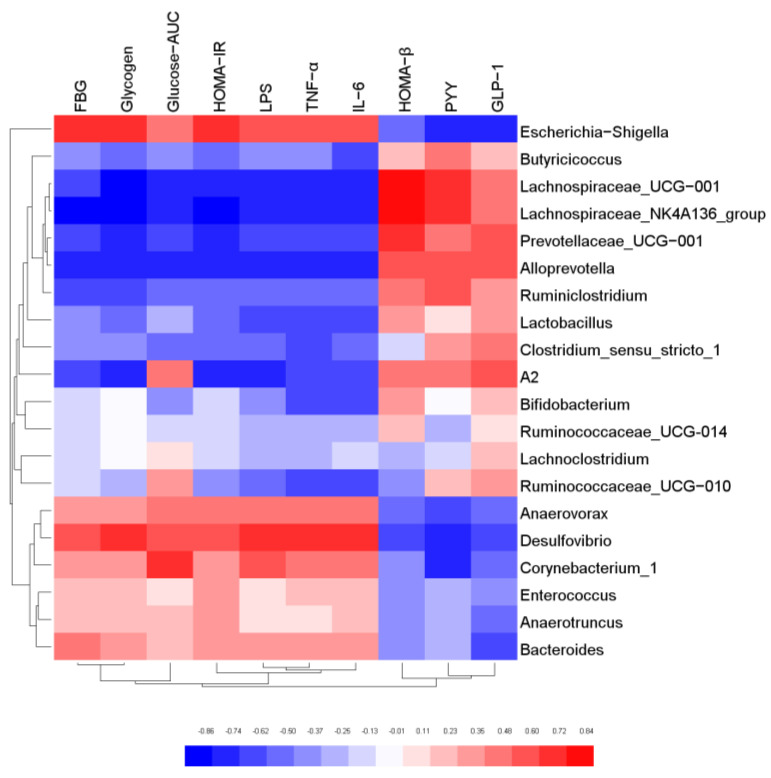
Correlation analysis of key gut microbiota and physical and chemical indicators.

**Table 1 foods-11-00754-t001:** The results of product index.

Content (mg/mL)	KT	TS
Total acid	8.6 ± 0.87	ND **
Total sugar	18.73 ± 2.87	158 ± 1.67 **
Total phenol acid	1.39 ± 0.15	0.09 ± 0.02 **
Formic acid	0.25 ± 0.06	ND **
Acetic acid	3.03 ± 0.81	ND **
Butyric acid	0.42 ± 0.11	ND **
Citric acid	0.67 ± 0.21	ND **
Lactic acid	0.41 ± 0.13	ND **

All data presented as means ± SD, ** represent *p* < 0.01, compared with non-fermented tea soup, ND indicates not detected.

## Data Availability

All data presented in this study are available in the main body of the manuscript.

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
