# Peer review of "Kombucha Reduces Hyperglycemia in Type 2 Diabetes of Mice by Regulating Gut Microbiota and Its Metabolites"

_foods, 2022, doi:10.3390/foods11050754_

Round 1

Reviewer 1 Report

The paper shows evidence of Kambucha_one of functional drink_have good effect to reduce hyperglycemia in type 2 DM of mice through regulating the gut microbiota and its metabolites. Most methods and results are good and shows high impact that Kombucha have good effect to reduce hyperglycemia. 

Some points should be clarified:

  1. Line 23, the word diet therapy should be change to 'functional drink'. The term of functional food/drink much better than diet therapy since the scope of the paper is related to functional food.
  2. Line 46, word 'e' should change to 'a'.
  3.  In the introduction section, please add what kind of species bacteria involved (acetic-lactic bacteria, and yeast) in fermentation of kombucha. The information of species is very important to reflect what kind of bioactive component derived from its metabolism.
  4. Please a bit explanation regarding functional food or functional drink in the Introduction section.
  5. Line 81-83, is it common diet for experimental diet? Please add the references in term of the using of corn flour, bean cake, bran, multivitamin and minerals? What kind of bran that use in this study? Please add more detail.
  6. Line 102 and Table 1, please change 'total phenol' to 'total phenolic acid'.
  7. Related to lines 213-214; what is the meaning of 'transformation in the fermentation process'? How its mechanism? What kind of enzyme involved? How fermentation increase total acid, total phenolic content or others active compounds? Does fermentation produce new compounds? or fermentation just only digest from cell wall of tea?
  8. Figure 2: how to connect increase food intake (1B) but decrease of BW (1C)? Suggestion Fig 1C, it is better to add data percentage of decreasing!

Reviewer 2 Report

Dear Authors,

Your work on how Kombucha beverage reduces hyperglycaemia in type 2 diabetes disease mouse model shows a clear result on the beneficial effect.

I have some minor revisions:

  • Please added spaces after punctuations and parenthesis. Example in line 10. This is a common mistake and should be addressed.
  • “rpm” in line 91 should be “RPM”
  • AUC formula does not show the “+”, this is just a formatting error.
  • Organ index is in percentage, please remove “%” at the end of the formula and add in parenthesis after “organ index”
  • Litters in short should be always in capital “L”, example “mL”. Please see line 218.
  • Missing “.” at the end of line 232
  • Salmonella, Vibrio cholerae, shigella at line 246 should be in italic.
  • Line 394 and elsewhere, please change the incorrect term “flora” to address intestinal microbes. The correct term is “microbiota”. No plant community was described in this work.

Kind regards,

Reviewer 3 Report

This manuscript is interesting and advances in the domain have been achieved and the content of the paper is related to the scope of the journal but the authors should consider the below comments to improve their paper.

  1. Please insert a list of abbreviations and nomenclature which would be useful considering the notations and equations used in the paper. Also, the unit for the involved parameters needs to be added in accordance with the SI.
    2. The authors should reformulate the abstract in order to emphasize the novelty of the paper. Lines 9-10 from the abstract and lines 55-56 from the introduction section are overlapping. It should be reformulated or removed.
    3. Conclusion need to be revised and future perspectives should be defined and added.
